# End-Triassic mass extinction started by intrusive CAMP activity

J.H.F.L. Davies[1], A. Marzoli[2], H. Bertrand[3], N. Youbi[4,5], M. Ernesto[6] & U. Schaltegger[1]

The end-Triassic extinction is one of the Phanerozoic's largest mass extinctions. This extinction is typically attributed to climate change associated with degassing of basalt flows from the central Atlantic magmatic province (CAMP). However, recent work suggests that the earliest known CAMP basalts occur above the extinction horizon and that climatic and biotic changes began before the earliest known CAMP eruptions. Here we present new high-precision U-Pb ages from CAMP mafic intrusive units, showing that magmatic activity was occurring ~100 Kyr ago before the earliest known eruptions. We correlate the early magmatic activity with the onset of changes to the climatic and biotic records. We also report ages from sills in an organic rich sedimentary basin in Brazil that intrude synchronously with the extinction suggesting that degassing of these organics contributed to the climate change which drove the extinction. Our results indicate that the intrusive record from large igneous provinces may be more important for linking to mass extinctions than the eruptive record.

[1] Département des sciences de la Terre, Université de Genève, Genève 1205, Switzerland. [2] Dipartimento di Geoscienze, Universitá di Padova, Padova 35131, Italy. [3] Laboratoire de Géologie de Lyon, ENS de Lyon, Université Lyon 1, CNRS, UMR 5276, Lyon 69364, France. [4] Cadi Ayyad University, Faculty of Sciences-Semlalia, Department of Geology, Marrakech Box 28/5, Morocco. [5] Instituto Dom Luiz, Faculdade de Ciências, Universidade de Lisboa, Lisboa 1749-016, Portugal. [6] Departamento de Geofisica, Instituto Astronomico, Geofisico e Ciencias Atmosfericas, Universidade de São Paulo, Rua do Matão, 1226, São Paulo, CEP 05508-900, Brazil. Correspondence and requests for materials should be addressed to J.H.F.L.D. (email: joshua.davies@unige.ch).

The central Atlantic magmatic province (CAMP) is one of the world's most expansive large igneous provinces (LIP) with an estimated areal extent of ca. $10^7$ km$^2$. The province was emplaced as mafic dykes, sills, layered intrusives and basalt flows in rift basins of the Pangean supercontinent at $\sim 201$ Ma (Fig. 1; refs 1–4). High-precision U-Pb geochronology of the CAMP basalts in eastern North America and Morocco[1] has temporally linked the onset of CAMP with the end-Triassic mass extinction (ETE) as determined by zircon U-Pb geochronology from ashes bracketing the extinction interval in Peru[5]. The ETE, the third largest of the Phanerozoic extinctions, is identified in the marine record by the loss of Triassic ammonites and devastation to the scleractinian corals[6,7]. In the continental record it is recognized by a turnover in sporomorphs and early Mesozoic vertebrates as well as a $> 95\%$ turnover of megaflora[8–12]. The ETE is also associated with a global $\sim 3$–6‰ negative carbon isotope excursion, which is recorded in both the marine and terrestrial records allowing the two records to be correlated[2,3,5,13–15]. Geochronologically, the age for the carbon isotope excursion from the marine and continental records overlap, however, there is still some controversy over whether the carbon isotope excursion in the orbitally tuned strata of the Newark basin really marks the extinction.

During the extinction period, atmospheric $CO_2$ concentrations were thought to have increased up to four times the pre-extinction levels creating up to 3–4 °C of warming[9]. Increased atmospheric $pCO_2$ concentrations are indicative of ocean acidification suggesting that this may have been a marine

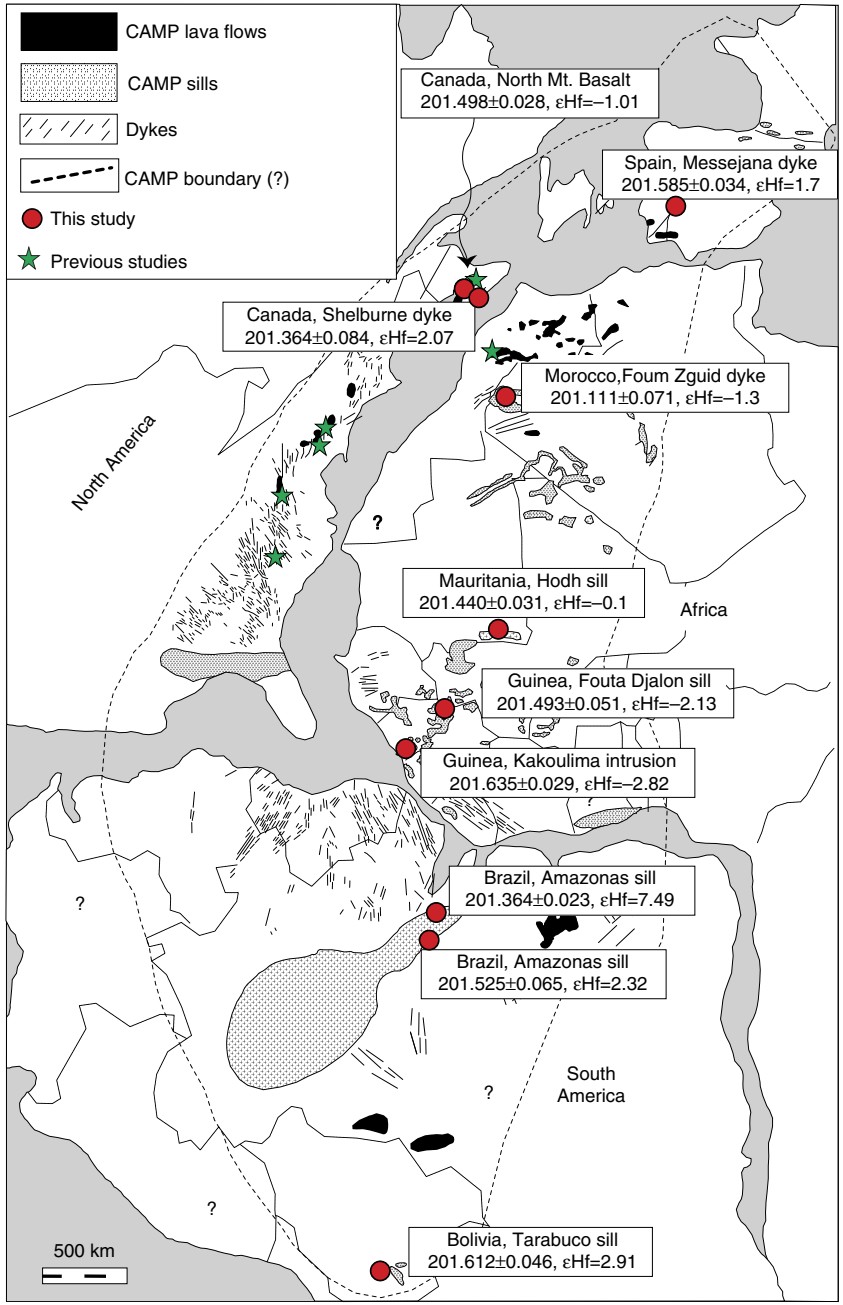

**Figure 1 | Map of the circum-Atlantic region during the Latest Triassic.** The positions of CAMP dykes, sills and flows are shown, along with locations of the dated samples, ages in Myr with associated 2 sigma uncertainties and εHf isotope values. Approximate locations of the present day continents and country borders are also shown. Previously dated samples are from ref. 5 and ref. 1. Map is modified after ref. 35.

extinction mechanism especially in relation to the scleractinian corals[16,17]. The temporal relationship between the CAMP and the ETE has led to the speculation that CAMP magmatism played a major role in the climate change and associated mass extinction. However, recent detailed carbon isotope stratigraphy on the sediments underlying the earliest known CAMP basalts in Morocco and North America has revealed that the flows were emplaced above the carbon isotope excursion marking the extinction horizon[2,3,18] questioning the causality of the link between the two.

Indirect evidence for CAMP activity before the earliest basalts has been discovered in different locations worldwide, for example, the identification of seismites below the ETE horizon in northern Europe, which have been interpreted as resulting from movement of magmas (presumed to be CAMP related) at depth[19]. Sediments from weathered mafic rocks are present below the first basalt flows in Morocco, which is interpreted as evidence for undiscovered basalts below the ETE[3]. Also, global sea level regression recorded in the Rhaetian sediments of UK, USA and Austria below the ETE has been interpreted as evidence for climatic cooling associated with early degassing of sulfur from CAMP magmas at depth[5,20]. Climatic change has also been suggested as an explanation for photic zone euxinia and increased ocean stratification in the northern Panthalassic Ocean before the ETE[21], as well as a loss of taxonomic richness in the bivalve and terrestrial vegetation records[11,12]. Currently, there is no direct evidence for CAMP activity before the earliest basalts, and coincident with or before the carbon isotope excursion that marks the ETE, therefore the link between the ETE and the eruption of CAMP basalts remains speculative.

The ETE is defined in the marine realm by a turnover in ammonite species and also by a marked negative carbon isotopic excursion[6,7]. This excursion has also been identified in the continental record in sediments at the base of CAMP lava flows in the Argana basin and Central High Atlas of Morocco and the Canadian Fundy basin in North America[2,3]. High precision geochronology from ashes bracketing the extinction interval and carbon isotope excursion in the marine record produces an age for the Triassic–Jurassic boundary of $201.36 \pm 0.17$ Myr and for the carbon isotopic shift marking the main phase of extinction of $201.51 \pm 0.15$ Myr ago (from ref. 5 recalculated by ref. 22). The age for the terrestrial ETE was calculated for the Newark basin as follows. First, the intrusion of the Palisades sill (dated at $201.520 \pm 0.034$ Ma) was assumed to belong to the same magma system as the Orange Mt. basaltic flow. This flow occurs $\sim 20$ m above a relatively modest carbon isotope excursion and a biologic turnover in the Newark basin, which has been interpreted to mark the extinction horizon. Considering that the sediments in the Newark basin have been orbitally tuned[23] and that the 20 m of sedimentary strata should correspond to a precession cycle, assuming no hiatus in sedimentary deposition, the age for the ETE in the Newark basin is $201.564 \pm 0.015$ Myr from ref. 1. This continental ETE age overlaps with the age for the extinction in the marine record.

However, despite the age overlap between the marine and terrestrial records, the biostratigraphic definition of the ETE in continental basins is strongly debated. In particular, the definition of the ETE in the Newark basins is based on a pollen and spore turnover[24] that is questioned by other palynologic and paleontological studies[25,26]. Further confusing the continental record, are suggestions that there may be a hiatus in sedimentation under the Orange Mountain basalt in the Newark basin (e.g. refs 2,22,27–29), which would invalidate the orbitally tuned age for the ETE. Above the Orange Mountain basalt, the floating chronology from the astronomically tuned Newark sediments matches well with the U-Pb ages[1] suggesting a

continuous sedimentary sequence above the basalts. Since the ages for the extinction horizon overlap between the marine and continental records we cautiously use the age of 201.564 Myr as an estimate of the age of the extinction, and acknowledge that this exact age may be refined in the future by further work in other continental basins straddling the end Triassic extinction.

Our new dates conclusively show, for the first time, that mafic magmatic activity associated with the CAMP was occurring $\sim 100$ kyr before the eruption of the previously dated Moroccan and North American basalts. We also show that shallow sills in the $\sim 1,000,000$ km$^2$ organic-rich Amazonas sedimentary basin of Brazil intrude simultaneously with the extinction, suggesting that degassing from organics in this basin may have contributed to the climatic impact of the CAMP, and consequently the ETE.

## Results

**U-Pb geochronology.** Here we present high precision U-Pb ages for zircon ($ZrSiO_4$) and baddeleyite ($ZrO_2$) from mafic intrusive rocks over four continents in eight locations throughout the CAMP (Fig. 1) to better understand the relationship between the CAMP and the ETE and to search for a possible CAMP pre-eruptive history (at least in relation to the exposed sections in North America and Morocco). The relative timing of CAMP magmas has traditionally been determined by comparison with the geochemical signatures of the Moroccan basalt flows[27]. However the geochemical chronology breaks down when considering the intrusive units that we targeted in this study (Supplementary Fig. 3). Intrusive rocks were chosen in order to circumvent problems associated with stratigraphic correlations between the marine record and the two continental records in Morocco and North America. All of the samples have been previously characterized geochemically and shown to be part of the CAMP (Supplementary Information), we also provide Hf isotope data from the dated minerals to reinforce the geochemical relationships (Fig. 2).

Our age of $201.635 \pm 0.029$ Myr for the Kakoulima layered mafic intrusion in Guinea is the oldest age for CAMP magmatism. It is $150 \pm 38$ kyr older than the North Mountain basalt (NMB) and $71 \pm 61$ kyr older than the Amelal sill both of which were previously the oldest dated CAMP units and geochemical correlatives with the intermediate basalts in Morocco (Fig. 2). After the initial magmatism recorded in Guinea, the Messejana dyke ($201.585 \pm 0.034$ Myr), Tarabuco sill ($201.612 \pm 0.046$ Myr), Fouta Djalon sill ($201.493 \pm 0.051$ Myr), Hodh sill ($201.440 \pm 0.03$ Myr) and the Amazonas (low Ti) sills ($201.525 \pm 0.065$ Myr) all intrude synchronously with the Palisades sill, York Haven, Rapidan, NMB and Amelal sill[1]. Following these early events, the Amazonas high Ti sill ($201.364 \pm 0.023$ Myr) and the Shelburne dyke ($201.336 \pm 0.084$ Myr) are synchronous with the Preakness basalt and the Rossville intrusive[1]. The Foum Zguid dyke ($201.111 \pm 0.071$ Myr) occurs $\sim 200$ kyr afterward the initial magmatism and just before the Butner intrusive[1], which yields the youngest recorded U-Pb age for the CAMP. Magmatism from the whole province shows remarkable temporal and isotopic consistency over $\sim 8,000$ km (Fig. 2).

**Hf isotopes.** The Hf isotopic compositions of the dated zircon and baddeleyite grains were determined in order to better understand the source of the CAMP magmas and to ensure that the zircon analyses used in age calculations are all from the same source (that is, not xenocrystic). Individual Hf isotopic analyses from each sample were averaged to create a mean value for comparison with other CAMP magmas from this study. All of the magmas have εHf values $\sim 0 \pm 2$ apart from the high Ti sample

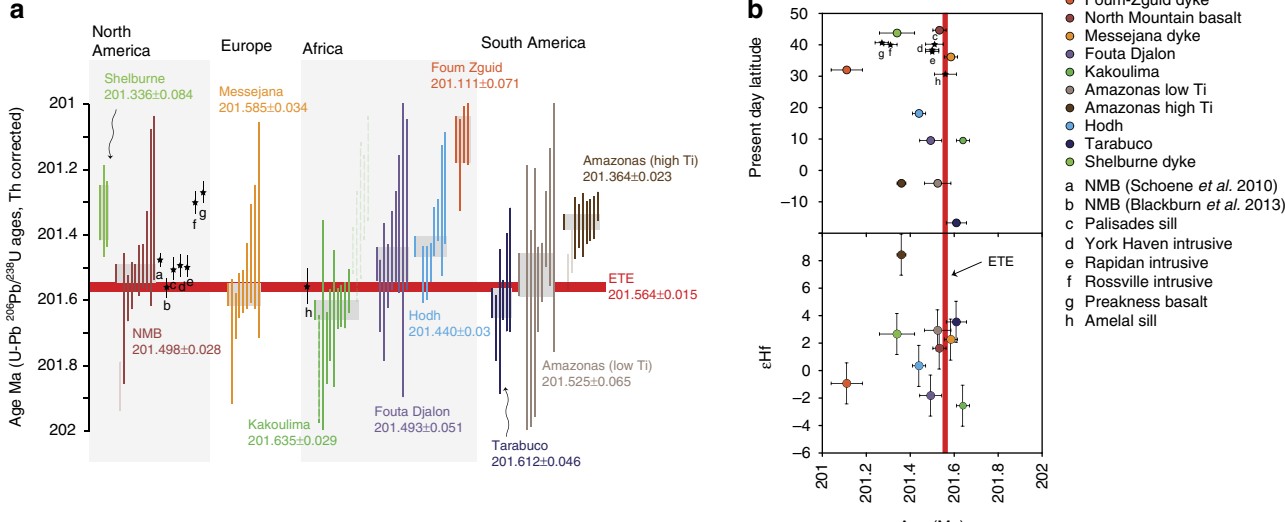

**Figure 2 | U-Pb geochronology and εHf isotope compilation for the CAMP showing spatial and temporal variations.** (**a**) Vertical coloured bars represent $^{238}U$-$^{206}Pb$ Th-corrected ages for single zircon (filled bar) and baddeleyite (dashed bar) crystals. The length of the bar reflects the 2σ analytical uncertainties, and the grey horizontal bar reflects the weighted mean age calculated for the sample, with the thickness of the bar reflecting the 2σ uncertainty on the weighted mean age. The red line represents the ETE from ref. 1, and the ages from that study, and ref. 5 are also shown. Note that our NMB sample overlaps in age, within uncertainty with both the ref. 1 and ref. 5 ages, although these do not overlap with each other, the age of the NMB should be 201.498 ± 0.028 Myr, which is an average of the ages from refs 1,5 and reported here (Supplementary Fig. 2). All ages shown are without the tracer calibration and decay constant uncertainties, these need to be included when comparing the U-Pb ages to ages from other isotope systems. (**b**) Weighted mean age and average εHf data from the dated intrusive rocks. εHf data uncertainties are all ±1.6ε units which represents the external reproducibility of the standard (see Methods section).

from the Amazonas basin in Brazil, which has a more primitive εHf value of ∼+7. The broadly chondritic Hf isotopic composition of the low Ti CAMP samples (Fig. 2) suggests a similar magmatic origin and a relatively homogenous Hf isotopic signature over the entire length of the province. Other tracer isotope systems (Sr, Nd Pb and Os) have been previously used to investigate the source of the CAMP magmas, however these have been dominantly analysed by whole rock techniques and often show a large range in values suggesting variable amounts of crustal contamination or variable crustal input into the mantle source over the whole province (see refs 30–35). The consistency of the zircon data suggests that (1) either the resolution of the technique is not at the required level to observe contamination, or (2) the contamination did not significantly alter the Hf isotopic composition of the samples. At present with the current resolution we cannot distinguish between the different possibilities.

The high εHf values from the high Ti samples from the Amazonas basin in Brazil are consistent with other tracer isotope systems suggesting that those magmas were derived from a depleted source distinct from that of the low Ti samples. Our new geochronology and Hf data is the first to conclusively show that the high Ti magmas occurred after the low Ti magmas.

**Geochemistry of the CAMP intrusives and Moroccan basalts.** The relative timing of CAMP magmas has traditionally been determined by comparison with the geochemical signatures of the Moroccan basalt flows[18,27,36]. Therefore it is pertinent to determine how the geochemistry of the CAMP intrusive units compares with the Moroccan flows. It should, however, be noted that direct comparison of the geochemical composition of near-liquidus lava flow samples with coarse-grained and often cumulitic intrusives should be done with caution.

Here we show that the geochemistry from most of the intrusive units do not coincide with each Moroccan flow. Most of the

intrusive suites overlap with multiple fields from the Moroccan lavas, predominantly clustering around the Intermediate and Upper flows (Supplementary Fig. 3). The Kakoulima samples cover a La/Yb range encompassing all of the Moroccan units but with lower TiO₂ contents, which is likely related to the cumulate nature of these rocks rather than a source characteristic[30]. Therefore the elemental geochemistry of the intrusive magmas is difficult to use as a tool for relative chronology within the CAMP.

**Discussion**
Our new data allow us to determine more precisely the relationship between the ETE and the CAMP. Importantly, our age of the Kakoulima intrusive indicates that CAMP magmatism was occurring ∼100 kyr before the eruption of the basalt flows in Morocco and North America. Since these basaltic lava flows occur above the ETE carbon isotope excursion, the new ages presented here provide direct evidence for older CAMP activity synchronous with the extinction.

A compilation of proxy records across the ETE shows that the timing of early $\delta^{13}C$ excursions before the ETE in the Newark-Hartford, St Audrie's bay, Stenlille and Kennecott Point records coincide with or just predate the newly discovered earliest CAMP magmatism (Fig. 3). Also, seismite occurrences before the ETE[19], correlate with the early ages as do changes in the atmospheric and ocean chemistry; the onset of increases in atmospheric $CO_2$ suggested by leaf stomatal density data from Jamesonland, Greenland and also in Scania[9]; increased stratification of the Panthalassic ocean recorded by the gammacerane index; reduction in bottom water oxidation recorded by homohopane increases; the onset of photic zone euxinia indicated by $C_{18-22}$ aryl isoprenoids and isorenieratane levels and changes to the N cycle indicated by $\delta^{15}N$ variations[21], bivalve biodiversity decreases[12]; and a dramatic decline in gymnosperm flora[11,12], all occur at the same time as the Kakoulima intrusive (Fig. 3).

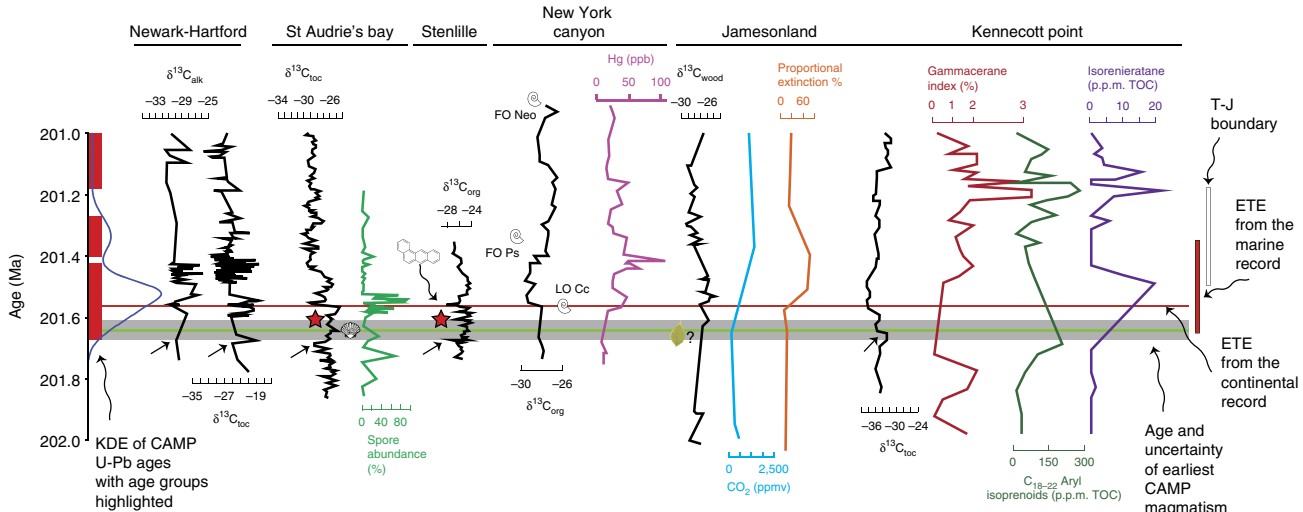

**Figure 3 | Detailed comparison and correlation between various proxy records around the ETE and the timing of CAMP intrusions.** A kernel density estimate of all high precision U-Pb CAMP ages from this study as well as those from ref. 1 and ref. 5 over a 1 Myr time period around the ETE are shown relative to various proxy records, whose age is calculated through astrochronology[1,2,65]. $\delta^{13}C_{toc}$ and $\delta^{13}C_{alk}$ records from the Newark and Hartford basins[2], $\delta^{13}C_{toc}$ and spore abundance (%) from St Audrie's bay[14,65], $\delta^{13}C_{toc}$ from the Stenlille basin[19], $\delta^{13}C_{toc}$ from New York Canyon[20], $\delta^{13}C_{wood}$ from Jamesonland combined with leaf stomatal density data from $CO_2$ p.p.m. reconstructions and proportional extinction % (ref. 10) and the $\delta^{13}C_{toc}$ and ocean redux indicators from ref. 21 from Kennecott point. The ETE in the continental record from ref. 1 is shown as a horizontal thin red line, the ETE period from the marine record from ref. 5 is shown by the box at the right of the diagram along with the Triassic–Jurassic boundary (T–J) from the same reference recalculated by ref. 22, both with associated 2σ uncertainty. The Kakoulima intrusion age and associated uncertainty is shown as a grey bar. Red stars indicate the position of seismites from ref. 19. Small arrows indicate the early carbon isotope excursions from ref. 3. LO is last occurrence, FO is first occurrence, Cc, *Choristoceras crickmayi*, Neo, *Neophylites*, Ps, *Psiloceras planorbis*. Broadleaf symbol marks the decline of complex terrestrial vegetation species in western Greenland[11,12], bivalve symbol represents start of bivalve taxenomic diversity decrease[12]. Benzoanthracene symbol and arrow indicate the identification of polycyclic aromatic hydrocarbons at the ETE horizon in the northern European basin[11]. Correlations are following ref. 2.

The coincidence of these events does not indicate causality, however, the geochronology can be used to tentatively link the events before the ETE to early CAMP magmatism. An open question is whether the early intrusive magmatism is associated with undiscovered basalt flows, which could degas during eruption, or whether degassing from the intrusives themselves could cause the disturbances to the proxy records. Degassing of magmas at depth has previously been suggested as a cause of global disturbance before the eruption of LIP flood basalts[37–39]. However, there is currently no known volatile rich magmatism (for example, kimberlites or carbonatites) at the same time as the Kakoulima intrusive which could suggest degassing of deep metasomatised rocks. We therefore, without further evidence, prefer not to speculate on early degassing of CAMP here.

After the onset of magmatism, marked by the Kakoulima intrusion, the majority of CAMP activity occurs coincident with, or just after the ETE[1,3,5,18]. The Messejana dyke in Spain, the Tarabuco mafic sill in Bolivia and importantly, mafic sills in the Amazon basin in Brazil occur synchronously with the 201.564 Myr age[1] for the extinction (Fig. 2). The Amazon basin covers ∼750,000 km² of NW Brazil and is broken into three sub-basins, the Amazonas (where the samples studied here are from) the Solimoes and the Acre. These basins have a thickness of up to 4.5 km (ref. 40), and contain some of the country's largest oil reserves, along with extensive Devonian shales and a Permo-Carboniferous evaporite sequences[41,42]. The intrusion of the CAMP sills into the Amazonas and Solimoes basins are known to have significantly altered the thermal history, causing oil generation and migration, as well as contact metamorphism of the organic and volatile rich sediments[42]. Polyaromatic hydrocarbons, which are likely created by burning and degassing of organics have been identified in the northern European basins[11] at the same time as the Amazonian gabbroic

sill intrusions suggesting that the two are linked. Degassing of Amazonian organic sediments would likely form pipe structures similar to those found in the Tunguska Basin in Russia where degassing of organic sediments was associated with the intrusion of sills from the Siberian Traps[43]. Pipes and associated metamorphic contact aureoles have not yet been identified in the Amazon basins, therefore it is difficult to estimate the volume and/or composition of gas released during degassing. However, the discovery that the sills intrude synchronously with the extinction quantitatively links CAMP magmatism with an important reservoir of volatiles, which would likely cause extensive atmospheric disturbance[44].

In conclusion, dated lava flows from northeastern North America[1] occurred after carbon isotope excursion marking the main phase of the end-Triassic extinction event and therefore cannot be the trigger. All of our new data instead show that degassing of intrusive magmas and/or surrounding volatile rich sediments (for example, from the Amazonas basin) before and contemporaneous with the extinction is the best candidate to trigger the climate change that caused the extinction.

## Methods

**Geochronology methods.** A total of 15 samples from across the CAMP were selected to undergo standard mineral separation procedures to extract zircon and baddeleyite crystals. All samples have been previously characterized geochemically (major elements, trace elements, ± isotopes) to indicate that they are part of the CAMP[30–36,45–49]. Ten of these samples (Fig. 1) contained zircon and baddeleyite that were subsequently dated, whereas the remaining five did not. The samples without dateable minerals were, AL1—Portuguese lava flow[34], B2—Kerforne dyke in Brittany, France[48], Caymar—gabbro dyke from Maringouins, Guyana[30], FERS 156—gabbro dyke from Fersiga Algeria[49], and MO11—Kaarta gabbro sill from western Mali[36].

In the datable samples, zircon and baddeleyite occur as trace phases in coarse mesostasis within the rocks and are often found in the presence of interstitial quartz and occasionally alkali-feldspar indicating that these igneous minerals are

crystalizing at a highly evolved, very late stage in the evolution of these mafic magmas (Supplementary Fig. 1). All reported ages are weighted mean $^{206}$Pb/$^{238}$U ages with associated 2σ uncertainties but without the systematic uncertainties associated with the tracer calibration and decay constants unless otherwise stated. All ages are corrected for initial $^{230}$Th disequilibria assuming a partition coefficient relationship (Th$_{(zircon/rock)}$/U$_{(zircon/rock)}$ of 0.2), for example, ref. 50 with the correction for disequilibria resulting in a ~90 kyr increase in the age of each grain. All zircons have undergone chemical abrasion treatment to remove the effects of Pb loss and subsequently no individual zircon ages younger than the weighted mean age are observed, validating our chemical abrasion procedures[51]. Unfortunately no such treatment is possible for baddeleyite, therefore many of the baddeleyite analyses are younger than the zircon from the same sample (Fig. 2 dashed lines) and are interpreted to have suffered Pb loss; these grains are not included in weighted mean calculations (Supplementary Data 1, Supplementary Fig. 1).

All extracted zircons were treated to remove areas of the crystal susceptible to Pb loss using chemical abrasion techniques; baddeleyite crystals were left untreated[52]. Zircon crystals were annealed in a muffle furnace for 48 h at 900 °C and subsequently inspected under a microscope and transferred into three ml Savillex beakers for chemical abrasion. Concentrated HF + trace HNO$_3$ was added to the beakers which were then placed into Parr bombs, and left in the oven at 180 °C for 15 h. After chemical abrasion, the remaining grain fragments (typically only fragments of grains were left after partial dissolution), fluxed overnight in 6 N HCl at 80 °C and ultra-sonically cleaned in ~3 N HNO$_3$ at least four times, each time with new clean acid. Zircon crystals were then loaded into 200 μl microcapsules, spiked with ~5 mg of EARTHTIME $^{202}$Pb + $^{205}$Pb + $^{233}$U + $^{235}$U tracer solution (calibration version 3.0, refs 53,54) and dissolved in ~70 μl concentrated HF and trace HNO$_3$ for 48 h at 210 °C in Parr bombs. Baddeleyite crystals were visually inspected using a binocular microscope and underwent the same ultrasonic cleaning procedure as zircon before being loaded into the microcapsules, spiked and dissolved. After dissolution, the samples were dried down and converted to chloride form and loaded onto columns filled with pre-cleaned anion exchange resin for U and Pb separation[55]. The remainder from the U and Pb separation (the wash) was collected for further Hf purification. Once separated, the U and Pb fractions were recombined in pre-cleaned 7 ml Savillex beakers and dried down with trace 0.035 N H$_3$PO$_4$ before loading on outgassed single Re-filaments with a Si-Gel emitter with a ratio of 0.06 Silic acid to 0.2 M HBr + 0.02 M H$_2$PO$_5$. U and Pb measurements were conducted on a Thermo TRITON thermal ionization mass spectrometer with Pb measured in dynamic mode on a MasCom secondary electron multiplier and U measured as UO$_2$ in static mode on Faraday cups equipped with 10$^{12}$ Ω resistors, assuming a $^{18}$O/$^{16}$O of 0.00205. Mass fractionation of Pb was corrected for using a $^{202}$Pb/$^{205}$Pb of 0.99924, and for U, $^{233}$U/$^{235}$U of 0.99506 was used along with a sample $^{238}$U/$^{235}$U of 137.818 ± 0.045 (2σ, ref. 56). The improved propagated uncertainty derived from online Pb fractionation corrections using the $^{202}$Pb + $^{205}$Pb tracer solution results in a ~0.01% increase in the precision of weighted mean ages relative to not using the double Pb tracer, and in this study that corresponds to ~ ± 20 kyr. All common Pb in the zircon and baddeleyite analyses was considered as blank and corrected using a long-term lab blank isotopic composition (see footnote in Supplementary Data 1). All U-Pb data was processed using Tripoli and Redux U-Pb software using the algorithms of ref. 57.

The ages presented here were calculated using the same EARTHTIME $^{202}$Pb-$^{205}$Pb-$^{233}$U-$^{235}$U spike as previous high precision studies[1] and therefore can be directly compared without propagating the uncertainties from the spike isotopic composition or the uranium decay constant, however, weighted mean ages with X/Y/Z uncertainties relating to analytical uncertainty (X), analytical + tracer isotopic composition uncertainty (Y) and analytical + tracer isotopic composition uncertainty + decay constant uncertainty (Z) are presented in Supplementary Data 1. To prove equivalence between the studies, we dated the North Mountain basalt (NMB), which has also been dated by refs 1,5, and our age overlaps with the ages from both previous studies.

**Age of the north mountain basalt.** The NMB has been dated by numerous studies, most of which produce similar ages of ~201 Myr. These studies include high precision U-Pb methods[1,5] lower precision U-Pb[58], Ar-Ar[59,60] and K-Ar[61]. Of these dating techniques, only the high precision U-Pb studies are relevant here since the ~percent level uncertainties for the other techniques preclude direct comparison with this study. This study and those of refs 1,5 all used the same ET2535 spike during analysis, therefore the ages should be directly comparable. The age of 201.481 ± 0.021 Myr[5] (updated to the current spike isotopic composition by ref. 22) was from a mafic pegmatite in Mount Pleasant quarry in the East Ferry Member (EFM) of the NMB, which is the lowest unit of the basalt. The NMB_03 sample dated here at 201.523 ± 0.028 Ma is from the same pegmatite, and both of our ages overlap within uncertainty. Reference 1 dated two samples from the EFM, one from the Mount Pleasant quarry, another from the Westport drill hole on the Brier Island. The Mount Pleasant quarry, which has an age of 201.566 ± 0.033 Myr, overlapping with the NMB age reported here, but not that of ref. 5. The second sample dated by ref. 1 is from the Westport drill hole and has an age of 201.522 ± 0.064 Myr which overlaps with all ages. The older Mount Pleasant quarry age[1] potentially contains evidence for older grains, possibly inherited cores

or antecrysts (Supplementary Fig. 2) and therefore may have been dragged to a slightly older age. To investigate the possibility for inherited zircon crystals within the NMB, we dated a further sample from the EFM, collected close to the balancing rock on Long Island, Nova Scotia (sample SU1201), this sample was homogeneous medium grained basalt with no pegmatite lenses. Zircon from SU1201 contains individual zircon crystal ages of up to 202.37 ± 0.41 Myr, which are interpreted as reflecting mixing between an inherited component and igneous zircon. These older zircon ages have relatively low radiogenic to common Pb ratios (Pb*/Pbc of <40) and are therefore susceptible to inheritance from partially dissolved older zircon fragments. CL images of zircon separates from SU1201 do not show obvious cores therefore the inherited zircon fragments are assumed to be at the micron scale or not in the plane transected during mounting and polishing. Contamination of the NMB magmas is also suggested by oxygen isotopes. Pyroxene δ$^{18}$O values of 10‰ from the Mount Pleasant quarry have been interpreted as reflecting contamination of mantle-derived magmas by continental crust[62].

We suggest that the best estimate for the age of the NMB as 201.498 ± 0.028 Myr, which is a weighted mean of the NMB_03 age from this study, the age from ref. 5 and also the Westport drill hole age from ref. 1 (Supplementary Fig. 2).

**Hf isotope methods.** Zircon from mafic rocks typically contains high trace element contents since it is one of the only phases preferentially partitioning heavy rare earth elements. The high trace element contents results in high Yb/Hf ratios (Supplementary Data 2) and for this reason, measuring the Hf isotopic composition on the washes from U-Pb chemistry[63] is not possible without purification. We used a simplified cation column chemistry procedure (similar to ref. 64) to purify the U-Pb washes so that they can be measured precisely on an MC-ICP-MS without isobaric interferences on $^{176}$Hf from Lu and Yb. Washes from the U-Pb column chemistry were collected in cleaned TPX beakers and dried down. They were then re-dissolved in an acid mix composed of 1 N HCl and 0.1 N HF and added to cation exchange columns which had been pre-cleaned with multiple washes of 6 N HCl, 29 N HF and water. After passing the U-Pb washes through the column, they were dried down and re-dissolved in 2% HNO$_3$ plus trace HF for MC-ICP-MS analysis.

Solution Hf isotope analyses were performed at the University of Geneva using a Thermo Neptune plus MC-ICP-MS equipped with nickel cones (Thermo 'X series') under dry plasma conditions using a Cetac 'Aridus 2' desolvation unit and a Teflon nebulizer (~50 ml min$^{-1}$ uptake rate) following the procedure of ref. 63. The nine Faraday cups were configured to measure the following masses at low mass resolution (m/_m ~450): 172 (L4), 173 (L3), 175 (L2), 176 (L1), 177 (C), 178 (H1), 179 (H2), 180 (H3), 181 (H4). The Plešovice and Temora zircon standards were run throughout the campaign at a Hf concentration of ~10 ppb to check for accuracy and reproducibility.

Many of the zircon grains measured in this study were small resulting in low concentrations of Hf, therefore Hf isotope measurements we employed following the short acquisition time measurements for the samples and sample-standard bracketing procedures outlined by ref. 63 with more concentrated Plešovice analysis measured over a longer time to accurately determine the mass bias of the sample by standard bracketing. Sample measurements were made using a 50 × 1 s acquisition time and Plešovice measurements utilized a 60 × 3 s acquisition. Yb and Lu signals were monitored to check for isobaric interferences but the cleanup column chemistry procedure was effective in reducing Yb and Lu to background levels. Correction for $^{176}$Hf in-growth due to $^{176}$Lu decay was corrected using a Lu/Hf of 0.001 which is similar to the Lu/Hf of the Temora zircon, changing this value by up to one order of magnitude results in an epsilon unit change of ~0.09, which is much less than the external reproducibility of the Plešovice zircon (~1.6 epsilon units), which was used as the uncertainty for all analysis. All Hf data from the samples and standards is provided in Supplementary Data 3.

**Data availability.** The data shown and discussed in this paper is presented in full in the Supplementary Material.

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

## Acknowledgements

This study was supported by the Swiss National Science Foundation (project number 162341). Partial funding was also provided by the following collaborative research projects: (i) PICS, CNRS (France)-CNRST (Morocco) to H.B. and N.Y. and (ii) CNRi (Italy)-CNRST (Morocco) to A.M. and N.Y. and we also acknowledge financial support from Padova University, CPDA132295/13, and from PRIN 20158A9CBM to A.M.

## Author contributions

U.S. and A.M. devised the project, J.H.F.L.D. collected the data, all authors contributed to the ideas and writing of the manuscript.

## Additional information

**Competing interests:** The authors declare no competing financial interests.

