## [Peer Review File · Nature Communications]

Reviewers' Comments:

Reviewer #1 (Remarks to the Author)

Davies et al present new U-Pb radioisotopic data from CAMP intrusive units that claim to show that magmatic activity started ~100 kya before the earliest known eruptions. The authors then attempt to correlate early magmatic activity with the onset of changes to the climatic and biotic records.

The conclusion that the intrusives predate the basalts requires one to believe that their claimed analytical precision can be used as a basis to define a geological accuracy. I see two problems with this:

1. There is mounting evidence that zircon residence times are usually over 10 ky, and sometimes well over 100 ky. Therefore, analytical uncertainties on the +/-10 ky scale such as the authors report do not mean much geologically. The results are much more ambiguous if the actual geological uncertainty is +/- 100 ky.
2. The reported low uncertainties partly derive from calculating weighted means from a group of individual crystal ages. But if you examine these ages, they do not cluster around a mean. As several recent papers have pointed out (Schoene, Mundil, etc.), in these cases, a weighted mean approach is probably inappropriate (as long as Pb loss is not the cause of the age spread). In these cases, the issue of zircon residence times indicates that it is likely that the youngest ages in each sample are closest to the actual intrusion/eruption age. Schoene et al. 2010 (Geology) and Schoene et al. 2015 (Science) are two examples of papers that uses the youngest crystal age rather than a weighted mean.

Another key issue is whether the extinctions actually predate the oldest lavas. Figure 3 is quite unconvincing in that regard. In fact, this figure shows the extinctions at the oldest lava if connected to the Blackburn et al paper. But in any case, the authors' data is shown way before the extinctions, in fact 100 ky before.

In sum, the claims are undermined by a lack of critical consideration of their data. The main conclusions of the paper are not supported if the true geological uncertainties of the age data cannot resolve a difference between the intrusives and basalts. Even if the dates are correct, the emplacement of those sills had no observable effect on the biota - direct evidence for extinctions being coincident with the igneous activity (Figure 3) is unconvincing at this time.

Reviewer #2 (Remarks to the Author)

Davies and his co-authors present a suite of world class high-precision geochronological data from intrusions near the Triassic Jurassic boundary that suggest, that the End Triassic mass extinction (suggested to be at 201.564 ± 0.015 Ma by Blackburn et al., 2013) may have been triggered by environmental change caused by the intrusion related release of greenhouse gases. Whereas previous studies seem to suggest that the main CAMP related basaltic effusive volcanism is mostly postdating the extinction event, the new high-precision ages that are presented here coincide with or predate the extinction event, which is the basis for this new hypothesis. Similar mechanisms have been suggested for other extinction events, although as yet geochronological data supporting this hypothesis are scarce and from that perspective alone this study is worthy of publication in a journal that reaches a wide audience.

A few issues need reconsideration which mainly affects the interpretation of the presented data. For each intrusive unit, the ages are presented as weighted mean age of a subset of the data set which, although many studies choose this approach, is a somewhat arbitrary *modus operandi*. This

approach is typically applied to extract a geologically “meaningful” age from complex data sets, in many cases an eruption age for volcanic events. In the case of the presented data, however, there is no real motivation to capture a specific event other than the duration of the intrusive event, which is likely represented by the entire data set since zircon crystallization presumably records a considerable portion of the intrusive history. The complexity of U-Pb zircon data sets from intrusions (in the sense of dispersed individual ages recording a protracted event) is now well known and the presented data are no exception, in that many of the presented data sets scatter in excess of the analytical uncertainty and suggest that intrusive activity lasted up to a few 100 kyr in each case. Rather than presenting somewhat arbitrary weighted mean ages it is therefore probably more useful to present the ages in a cumulative approach like the presented in the attached graph. It illustrates that intrusive activity of all studied intrusions commences at 202 Ma, and peak activity seems to be bimodal at 201.526 Ma (within 25 kyr of the purported extinction event), and at 201.376 Ma, and then rapidly subsiding. The illustration clearly shows that a significant portion of the magmatic activity already occurs prior to the extinction event. To capture a fully realistic picture, more data of this quality from a greater number of intrusions are needed but the presented data set is quite extensive (in fact rather unique in terms of volume for a U-Pb TIMS study).

The other issue is of course the “true” age of the mass extinction which is somewhat ambiguous and, unfortunately, only discussed in the supplemental materials. Another recent study suggests that the marine extinction occurs at 201.36 ± 0.17 Ma rather than the above quoted age of the mass extinction. The latter age coincides (probably fortuitously) with the younger peak of the probability plot. A brief discussion of the ambiguity of the age of the extinction should be included in the main text.

Unfortunately, a discussion of the estimated magnitude of the impact of released gases on the environment of the early intrusions is missing and relevant studies addressing this are not even quoted (e.g. Ganino and Arndt, 2009, *Geology*) are not cited. This topic should be briefly discussed. Neither of these shortcomings are meant to take anything away from the exceptional quality of the presented data.

Reviewer #3 (Remarks to the Author)

This is a neat study that reaffirms the link between the onset of LIP volcanism and mass extinction which is well established. It attempts to be more novel than this by highlighting that a recent study of the Moroccan basalts show them to be a little young to be implicated in the mass extinction but this does not invalidate the general tight linkage between these phenomena – the CAMP was vast there could well be older basalts elsewhere in the region. So, despite the claims here about “questioning causality” the link is excellent, not “speculative”, whatever ref. 3 may try to claim. The much clearer link is between the onset of mass extinctions and LIP volcanism that is now seen for many events (e.g. the Middle Permian is well shown).

I also wonder whether it is possible to separate the effects on intrusives and extrusives, they will surely be too closely spaced in time to distinguish - at least for an event that was 200 myrs ago. Also, can huge volume, high level sills remain below the surface for ~100 000 years, causing baking and explosive eruptions of thermogenic and volcanigenic gases, without erupting any lava? This is what this study claims but it seems unlikely, and given the likelihood that surface basalts will weather and erode it is hard to test in the geological record.

Reviewers' comments:

Reviewer #1 (Remarks to the Author):

Davies et al present new U-Pb radioisotopic data from CAMP intrusive units that claim to show that magmatic activity started ~100 kya before the earliest known eruptions. The authors then attempt to correlate early magmatic activity with the onset of changes to the climatic and biotic records.

The conclusion that the intrusives predate the basalts requires one to believe that their claimed analytical precision can be used as a basis to define a geological accuracy. I see two problems with this:

1. There is mounting evidence that zircon residence times are usually over 10 ky, and sometimes well over 100 ky. Therefore, analytical uncertainties on the +/-10 ky scale such as the authors report do not mean much geologically. The results are much more ambiguous if the actual geological uncertainty is +/- 100 ky.

COMMENT

The first part of the first comment is concerned with the calculation of weighted mean ages at ~30 ky precision when, in other studies zircon (from very different rock types) is shown to crystallize over a protracted time period much greater than the precision presented in this study. This concern is dealt with in our response to the reviewers' second point below. The second part of the comment questions whether our analytical uncertainties are reasonable. All sources of analytical uncertainty (measurement related, tracer isotopic composition, blank isotopic composition, decay constant uncertainties) are fully propagated and presented with appropriate significant digits in the data table, and the tracer isotopic composition along with the formulas for error propagation and age calculation used in this study have been previously published (Condon et al. 2015; McLean et al., 2015). However, we acknowledge that there may be some confusion with the way the ages are presented and since we compare our results to other U-Pb ages from the CAMP using the same tracer solution, we do not include the uncertainties associated with the tracer calibration and decay constant since these are equal between studies and can be cancelled out. However when the ages presented here, and also those from other U-Pb studies are compared with ages from other isotope systems (Ar-Ar for example) the full uncertainty needs to be used. Our weighted mean ages with the X/Y/Z uncertainties after Schoene and Bowring (2006) (with X representing the analytical uncertainty, Y – analytical + tracer uncertainty and Z – the Y uncertainty + decay constant uncertainty) are given in the data table.

ACTION

We have added a sentence to caption of Fig 2 indicating that the uncertainties for the weighted mean ages are without tracer and decay constant errors. We have also added a sentence to the analytical techniques section of the supplementary information indicating that ages with the different levels of analytical uncertainty can be found in the data table.

2. The reported low uncertainties partly derive from calculating weighted means from a group of individual crystal ages. But if you examine these ages, they do not cluster around a mean. As several recent papers have pointed out (Schoene, Mundil, etc.), in these cases, a weighted mean approach is probably inappropriate (as long as Pb loss is not the cause of the age spread). In these cases, the issue of zircon residence times indicates that it is likely that the youngest ages in each sample are closest to the actual intrusion/eruption age. Schoene et al. 2010 (Geology) and Schoene et al. 2015 (Science) are two examples of papers that uses the youngest crystal age rather than a weighted mean.

COMMENT

We acknowledge the reviewers comment that most recent high precision zircon studies have shown that zircon residence times in felsic magmas and also the presence of zircon antecrysts from pulsed magmatic growth of felsic plutons preclude the use of weighted mean ages to calculate the age of a sample. However, all of the ages in our study are from zircon and baddeleyite crystalizing from mafic intrusive magmas where residence and inheritance are on the whole not an issue. We refer to the study by Schoene et al. (2010) to illustrate our point. The North Mountain basalt age from Schoene et al. (2010) (also dated in our study) was a weighted mean age from 32 zircon crystals, all of which were overlapping in age producing a weighted mean age uncertainty of ~20 Ka (only including the X component of uncertainty, see above) and an MSWD of 1.2. Zircon is a very late crystalizing phase in mafic rocks and is typically under-saturated in mafic melts meaning that inheritance and antecrystic zircon is rare. However, the ashes that were also dated by Schoene et al. (2010) that bracket the carbon isotope anomaly and extinction horizon, were from felsic volcanics and have much more complex age distributions. These ash ages are interpreted to represent “protracted growth of zircon in the [felsic] magmatic system and/or the incorporation of xenocrystic zircon” and the best estimate for the age of the ash is obtained from the youngest zircon. These two examples indicate the difference in zircon age distributions from felsic volcanics (and also felsic intrusive rocks) and mafic rocks, and how the ages should be treated. Also, to further illustrate the point, the ages in Schoene et al. (2015), which is referenced by the reviewer as supporting the case for using the youngest zircon age rather than weighted mean ages, are all from felsic volcanic ashes interbedded with the Deccan basalts rather than from the basalts themselves. There is considerable complexity in these ages, and we are confident that had Schoene et al. 2015 found zircon in the Deccan basalts, they would have preferred to use those rather than the zircons from the ashes.

ACTION

Despite the difference between zircon in felsic and mafic systems, we acknowledge that the majority of Nature Communications readers will not be fully familiar with these differences and therefore we have made some changes to the text so that readers should not misinterpret or prematurely reject our conclusions. We use the word “basalt” 22 times in the text to indicate that we are dealing with mafic rocks, we have now added the prefix ‘mafic’ when referring to the intrusive rocks we are dating to ensure that readers do not consider the system to be the same as felsic intrusives. In the supplementary information we also provide thin section evidence that the zircon and baddeleyite are crystalizing in late stage evolved melt pockets that contain quartz and feldspar, which suggests that zircon residence or inheritance is unlikely to be an issue in our

samples. In the data table we also provide MSWD values for our weighted mean ages so that readers can determine the spread in ages around the mean (almost all of our weighted mean ages have MSWD values that indicate the spread around the mean is attributable to analytical uncertainty). We also provide Hf isotope data for most of the dated zircon and baddeleyite crystals that shows that with individual samples, the grains crystallized from melt with the same isotopic composition, this would be unlikely if the grains were antecrysts crystallized in a melt different from CAMP basalt.

Another key issue is whether the extinctions actually predate the oldest lavas. Figure 3 is quite unconvincing in that regard. In fact, this figure shows the extinctions at the oldest lava if connected to the Blackburn et al paper. But in any case, the authors' data is shown way before the extinctions, in fact 100 ky before.

COMMENT

In the manuscript we stress that the age relationships between the extinction, which is marked by the carbon isotope excursion in the terrestrial record, and the basalts, are based on stratigraphic constraints. Therefore, if the age of a basalt, overlying the carbon isotope excursion has an age overlapping with the extinction age, the dating technique is not precise enough to determine the age difference. However, as the reviewer notes, our oldest age is much older than the proposed age for the extinction and is resolvably different given the analytical precision we possess.

In sum, the claims are undermined by a lack of critical consideration of their data. The main conclusions of the paper are not supported if the true geological uncertainties of the age data cannot resolve a difference between the intrusives and basalts. Even if the dates are correct, the emplacement of those sills had no observable effect on the biota - direct evidence for extinctions being coincident with the igneous activity (Figure 3) is unconvincing at this time.

Reviewer #2 (Remarks to the Author):

Davies and his co-authors present a suite of world class high-precision geochronological data from intrusions near the Triassic Jurassic boundary that suggest, that the End Triassic mass extinction (suggested to be at 201.564 ± 0.015 Ma by Blackburn et al., 2013) may have been triggered by environmental change caused by the intrusion related release of greenhouse gases. Whereas previous studies seem to suggest that the main CAMP related basaltic effusive volcanism is mostly postdating the extinction event, the new high-precision ages that are presented here coincide with or predate the extinction event, which is the basis for this new hypothesis. Similar mechanisms have been suggested for other extinction events, although as yet geochronological data supporting this hypothesis are scarce and from that perspective alone this study is worthy of publication in a journal that reaches a wide audience.

A few issues need reconsideration which mainly affects the interpretation of the presented data. For each intrusive unit, the ages are presented as weighted mean age of a subset of the data set which,

although many studies choose this approach, is a somewhat arbitrary *modus operandi*. This approach is typically applied to extract a geologically “meaningful” age from complex data sets, in many cases an eruption age for volcanic events. In the case of the presented data, however, there is no real motivation to capture a specific event other than the duration of the intrusive event, which is likely represented by the entire data set since zircon crystallization presumably records a considerable portion of the intrusive history. The complexity of U-Pb zircon data sets from intrusions (in the sense of dispersed individual ages recording a protracted event) is now well known and the presented data are no exception, in that many of the presented data sets scatter in excess of the analytical uncertainty and suggest that intrusive activity lasted up to a few 100 kyr in each case. Rather than presenting somewhat arbitrary weighted mean ages it is therefore probably more useful to present the ages in a cumulative approach like the presented in the attached graph. It illustrates that intrusive activity of all studied intrusions commences at 202 Ma, and peak activity seems to be bimodal at 201.526 Ma (within 25 kyr of the purported extinction event), and at 201.376 Ma, and then rapidly subsiding. The illustration clearly shows that a significant portion of the magmatic activity already occurs prior to the extinction event.

COMMENT

The main issue regarding the use of weighted mean ages is discussed in our reply to the first reviewer. We also acknowledge that there is some complexity with our age results, however, most of this complexity comes from dating baddeleyite grains which cannot be treated by chemical abrasion to remove the effects of secondary, post-crystallization Pb loss. Most of the baddeleyite crystals we dated are younger than the chemically abraded zircon, we attribute these younger ages to Pb loss. We have added thin section images (see above) that show igneous zircon and baddeleyite in the Foug Zguid dyke (the other samples show similar features). Zircon and baddeleyite are co-crystallizing in the image and therefore should have the same age. If the baddeleyite ages that are younger than the weighted mean zircon age are removed, the majority of the age complexity disappears and the zircons from a single sample overlap in age.

We agree with the reviewer that reporting the ages as using a cumulative approach is more useful than showing individual ages. We show a KDE of all of the U-Pb dated samples (from the current study, Blackburn et al. 2013 and Schoene et al. 2010) in Fig 3. We prefer to use a KDE of the sample ages rather than PDF of all of the zircon ages for two reasons, first the KDE attempts to represent the underlying distribution that the samples dated have been taken from rather than the PDF which just sums up the age and probability from each sample. Secondly, we feel that using the sample ages rather than zircon ages reduces the bias towards samples that have more zircon. However, despite these small quibbles using a KDE or a PDF on the sample ages or the zircons themselves produces essentially similar results. Figure 3 clearly shows that a significant proportion of the age probability is older than the extinction event. We actually did not notice that the second age peak is at 201.3 as noticed by the reviewer and that this age is identical to age of extinction recorded in the marine record, we acknowledge that this deserves some attention in the text, we respond further below.

To capture a fully realistic picture, more data of this quality from a greater number of intrusions are needed but the presented data set is quite extensive (in fact rather unique in terms of volume for a U-Pb TIMS study).

COMMENT

We agree that more ages are needed to fully constrain the nature of the CAMP

The other issue is of course the “true” age of the mass extinction which is somewhat ambiguous and, unfortunately, only discussed in the supplemental materials. Another recent study suggests that the marine extinction occurs at 201.36 ± 0.17 Ma rather than the above quoted age of the mass extinction. The latter age coincides (probably fortuitously) with the younger peak of the probability plot. A brief discussion of the ambiguity of the age of the extinction should be included in the main text.

COMMENT

We fully agree with this comment, however, the exact age of the ETE is somewhat complicated by the various methods used to determine it; in the marine record, the ETE is dated by U-Pb dating of ashes bracketing the ammonite turnover in Peru and correlating this to the ammonite turnover and carbon isotope excursion in the New York Canyon (Schoene et al. 2010). In the continental record the ETE is dated assuming that the sediments in the Newark basin are orbitally forced, and using this record to determine the age of the carbon isotope excursion. Further complications arise from the fact that the orbitally forced sediments in the Newark basin are tied to an absolute chronology through U-Pb dating of the basalt flows in the Newark and surrounding basins. These U-Pb dates were made using the same ET2535 tracer solution that was used in the Schoene et al. 2010 study, and also in this present study allowing the dates to be directly compared. However, the Schoene et al. 2010 study used an old version of the tracer isotope calibration, if the Schoene et al. 2010 ages are re-calculated to reflect the updated tracer calibration (e.g. Wotzlaw et al. 2014), the age for the ETE in the marine record becomes 201.5 ± 0.15 Ma, which overlaps with the age from Blackburn et al. 2013.

ACTION

We have mentioned the complexities regarding the exact extinction age in the main text, and also refer to the more detailed discussion in the supplementary information. We also add in Figure 3 the updated age of the ETE in the marine record, as well as the updated age of the Triassic-Jurassic boundary, and also add this information to the figure caption.

Unfortunately, a discussion of the estimated magnitude of the impact of released gases on the environment of the early intrusions is missing and relevant studies addressing this are not even quoted (e.g. Ganino and Arndt, 2009, *Geology*) are not cited. This topic should be briefly discussed. Neither of these shortcomings are meant to take anything away from the exceptional quality of the presented data.

COMMENT

We appreciate the reviewer’s comment and agree with it

ACTION

We include this reference in our section that mentions the effects of magma intrusion into sedimentary basins, which may subsequently degas. We intentionally held back on the degassing story since our data is mostly geochronological in nature (plus some isotopic tracing) and therefore we felt that discussing in detail different degassing scenarios beyond the scope of the present work.

Reviewer #3 (Remarks to the Author):

This is a neat study that reaffirms the link between the onset of LIP volcanism and mass extinction which is well established. It attempts to be more novel than this by highlighting that a recent study of the Moroccan basalts show them to be a little young to be implicated in the mass extinction but this does not invalidate the general tight linkage between these phenomena – the CAMP was vast there could well be older basalts elsewhere in the region. So, despite the claims here about “questioning causality” the link is excellent, not “speculative”, whatever ref. 3 may try to claim. The much clearer link is between the onset of mass extinctions and LIP volcanism that is now seen for many events (e.g. the Middle Permian is well shown).

COMMENT

We acknowledge the excellent link between large igneous provinces and mass extinctions, however, in the case of the CAMP, the oldest known basaltic lava flows do appear above the carbon isotope excursion that is interpreted to reflect the mass extinction (as noted by Dal Corso et al. 2014 and Deenen et al., 2010). We acknowledge that this doesn't mean that there are not older undiscovered basalts elsewhere in the CAMP – it is such a huge province that it would be naïve to say as such.

I also wonder whether it is possible to separate the effects on intrusives and extrusives, they will surely be too closely spaced in time to distinguish - at least for an event that was 200 myrs ago. Also, can huge volume, high level sills remain below the surface for ~100 000 years, causing baking and explosive eruptions of thermogenic and volcanogenic gases, without erupting any lava? This is what this study claims but it seems unlikely, and given the likelihood that surface basalts will weather and erode it is hard to test in the geological record.

COMMENT

We do not claim that the magmas forming the intrusions in Brazil do not erupt somewhere (in fact lava flows occur in the Parnaiba basin (Merle et al., 2011), but we were unfortunately not able to extract zircon from those flows (as indicated in the supplementary methods section). We just indicate that degassing from these sediments would contribute to the climatic impact of the CAMP, and this is especially significant since the sills maybe the same age as the extinction event.

References

- Blackburn, T. J. *et al.* Zircon U-Pb Geochronology Links the End-Triassic Extinction with the Central Atlantic Magmatic Province. *Science* **340**, 941–945 (2013).
- Condon, D. J., Schoene, B., McLean, N. M., Bowring, S. A. & Parrish, R. R. Metrology and traceability of U–Pb isotope dilution geochronology (EARTHTIME Tracer Calibration Part I). *Geochimica et Cosmochimica Acta* **164**, 464–480 (2015).
- Dal Corso, J. *et al.* The dawn of CAMP volcanism and its bearing on the end-Triassic carbon cycle disruption. *Journal of the Geological Society* **171**, 153–164 (2014).
- McLean, N. M., Condon, D. J., Schoene, B. & Bowring, S. A. Evaluating uncertainties in the calibration of isotopic reference materials and multi-element isotopic tracers (EARTHTIME Tracer Calibration Part II). *Geochimica et Cosmochimica Acta* **164**, 481–501 (2015).
- Merle, R. *et al.* $^{40}\text{Ar}/^{39}\text{Ar}$ ages and Sr–Nd–Pb–Os geochemistry of CAMP tholeiites from Western Maranhão basin (NE Brazil). *Lithos* **122**, 137–151 (2011).
- Schoene, B., Bowring, S.A., U–Pb systematics of the McClure Mountain syenite: thermochronological constraints on the age of the Ar-40/Ar-39 standard MMhb. *Contributions to Mineralogy and Petrology*, **151 (5)**, 615–630 (2006)
- Schoene, B., Guex, J., Bartolini, A., Schaltegger, U. & Blackburn, T. J. Correlating the end-Triassic mass extinction and flood basalt volcanism at the 100 ka level. *Geology* **38**, 387–390 (2010).
- Schoene, B., *et al.* U-Pb geochronology of the Deccan Traps and relation to the end-Cretaceous mass extinction. *Science* **347 (6218)**, 182–184 (2015)
- Wotzlaw, J.-F. *et al.* Towards accurate numerical calibration of the Late Triassic: High-precision U-Pb geochronology constraints on the duration of the Rhaetian. *Geology* **42**, 571–574 (2014).

Reviewers' Comments:

Reviewer #2:

Remarks to the Author:

I am overall ok with the modifications by the authors.

Reviewer #3:

Remarks to the Author:

There is a lot of work in this manuscript and it does a good job of showing the precise synchrony between CAMP intrusives and the main extinction losses of the end-Triassic mass extinction. Based on their data, they are certainly justified in making the claim that it is the intrusives (and associated volatile release either volcanogenic and/or thermogenic), that are responsible for the environmental changes and not the flood basalt eruptions. These are slightly too young as has been known for a while. There is always the problem that basalt flows are much more vulnerable to erosion and so may be lost or less readily discovered, but they can only go with what they find.

There are a few minor issues, mostly with the phraseology:

58: "could have lead to" would be better than "are indicative"

67: "questioning the causality" – are there any references to support this "questioning"? Who has questioned the role of LIP volcanism for the TJ mass extinction?

129: "long before" is too vague, spell out how long before.

173: Ref 29 – this is incomplete – what sort of publication is this? A conference abstract? Not really true support for the idea of contact metamorphism.

190: should really be "and/or" rather than just "or"

Replies to reviewers

Reviewer #2 (Remarks to the Author):

I am overall ok with the modifications by the authors.

Reply -
no need for response

Reviewer #3 (Remarks to the Author):

There is a lot of work in this manuscript and it does a good job of showing the precise synchrony between CAMP intrusives and the main extinction losses of the end-Triassic mass extinction. Based on their data, they are certainly justified in making the claim that it is the intrusives (and associated volatile release either volcanogenic and/or thermogenic), that are responsible for the environmental changes and not the flood basalt eruptions. These are slightly too young as has been known for a while. There is always the problem that basalt flows are much more vulnerable to erosion and so may be lost or less readily discovered, but they can only go with what they find.

There are a few minor issues, mostly with the phraseology:

58: “could have lead to” would be better than “are indicative”

Reply-
Changed

67: “questioning the causality” – are there any references to support this “questioning”? Who has questioned the role of LIP volcanism for the TJ mass extinction?

Reply –

We provide three references which show that the earliest known basalt flows occur above the carbon isotope excursion which is thought to mark the onset of the end-Triassic extinction on the continents. The observation that the earliest known basalt flows occur after the carbon isotope excursion suggests that degassing from the basalt flows was not the cause of the extinction.

129: “long before” is too vague, spell out how long before.

Reply –

We are now more specific about the amount of time before

173: Ref 29 – this is incomplete – what sort of publication is this? A conference abstract? Not really true support for the idea of contact metamorphism.

Reply –

We have provided an updated and complete reference to support our point

190: should really be “and/or” rather than just “or”

Reply –

Changed